# Prevalence and intensity of soil-transmitted helminth infections and associated risk factors among household heads living in the peri-urban areas of Jimma town, Oromia, Ethiopia: A community-based cross-sectional study

Ahmed Zeynudin[1]*, Teshome Degefa[1], Million Tesfaye[2], Sultan Suleman[3], Elias Ali Yesuf[4], Zuber Hajikelil[5], Solomon Ali[6], Khalide Azam[7], Abdusemed Husen[8], Jafer Yasin[9], Andreas Wieser[10,11,12]

1 Department of Medical Laboratory Sciences, Institute of Health, Jimma University, Jimma, Ethiopia, 2 Departments of Anesthesia, Institute of Health, Jimma University, Jimma, Ethiopia, 3 Departments of Pharmacy, Institute of Health, Jimma University, Jimma, Ethiopia, 4 Department of Health Policy and Management, Institute of Health, Jimma University, Jimma, Ethiopia, 5 Department of Medical Laboratory Sciences, Wolkite University, Wolkite, Ethiopia, 6 Saint Paul's Hospital Millennium Medical College, Addis Ababa, Ethiopia, 7 The East, Central and Southern African Health Community (ECSA-HC), Arusha, Tanzania, 8 Department of Oncology, Institute of Health, Jimma University, Jimma, Ethiopia, 9 Oda Hulle Primary Hospital, Jimma, Ethiopia, 10 Division of Infectious Diseases and Tropical Medicine, University Hospital, Ludwig-Maximilians-Universitat (LMU) Munich, Munich, Germany, 11 Department of Bacteriology, Max von Pettenkofer-Institute (LMU), Munich, Germany, 12 German Center for Infection Research (DZIF), Munich, Germany

* ahmed.zeynudin@ju.edu.et

## Abstract

### Background

Ethiopia has set national targets for eliminating soil-transmitted helminths (STH) as public health problems by 2020 and for breaking their transmission by 2025 using periodic mass treatment of children in endemic areas. However, the status of STH infection among the adults living in the same communities remains unknown. The aim of this study, therefore, was to determine the prevalence and intensity of STH infections and associated factors among the household heads in the peri-urban areas of Jimma town, Oromia, Ethiopia.

### Methods

A community-based cross-sectional study was conducted in five peri-urban kebeles (smallest administrative unit in Ethiopia) of Jimma town from May to July 2021. A semi-structured questionnaire was used to collect data on socio-demographic and predisposing factors. The Kato-Katz concentration technique was utilized to detect and quantify the STH in stool samples. Both bivariate and multivariate analyses were done. P-value <0.05 was considered statistically significant.

**Data Availability Statement:** All relevant data are within the paper and its Supporting Information files.

**Funding:** Initials of the authors who received the award = Ahmed Zeynudin Grant numbers awarded to the author = Not applicable The full name of the funder= The Bavarian State Government through CIH-LMU URL of funder website: https://www.cih.lmu.de the funders have no any role in the study design, data collection and analysis, decision to publish, or preparation of the manuscript.

**Competing interests:** The authors declare that they have no conflict of interest.

## Results

A total of 376 household heads (19.9% women and 80.1% men) from peri-urban areas were included in the study. The overall STH prevalence was 18.1% (95% CI: 14.6–22.1) with *A. lumbricoides* being the predominant species (11.4%) followed by *T. trichiura* (7.2%) and hookworm (2.1%). Most of the STH positive household heads had single infections (85.3%) and light-intensity infections (88.5%). Wealth status (AOR = 2.7; 95% CI: 1.31–5.50, P = 0.007), hand washing habits before meals (AOR = 7.07; 95% CI: 1.79–27.88, p = 0.005), fingernails status (AOR = 2.99; 95% CI: 1.59–5.65, p = 0.001), and toilet facility type (AOR = 2.06; 95% CI: 1.13–3.76, p = 0.017) were found to have statistically significant associations with the STH infection.

## Conclusion

The findings of this study showed a nearly moderate level of STH prevalence among household heads in the peri-urban community. This could serve as an important reservoir for reinfection of the treated children and other at-risk groups in the community.

## Introduction

Soil-transmitted helminths (STH) are among the most common and widely distributed infections, particularly in tropical and subtropical countries where poverty, inadequate sanitation and hygiene are common [1,2]. Globally, over 1.5 billion people are estimated to be infected with STH, leading to an estimated 3.3 million disability-adjusted life years [3–5]. *Ascaris lumbricoides*, *Trichuris trichiura*, and hookworms (*Necator americanus* and *Ancylostoma duodenale*) are the three main species of concern that infect humans with a global prevalence estimated to be 1.2 billion, 795 million, and 740 million people, respectively [6]. Sub-Saharan Africa [7], the Americas, India, China and East Asia are the most affected regions, predominantly carrying the highest burden of STH infections [2,4,8].

In sub-Saharan Africa (SSA), helminth infections account for approximately 85% of the neglected tropical diseases, with hookworm infection affecting almost half of the SSA's poorest communities [9]. Over the past decades, there are reports of considerable reductions in the intensity of STH infections through a combination of measures including preventive chemotherapy (PC), improved water supplies and sanitation, as well as hygiene education programs implemented in different countries in SSA [10,11]. Nevertheless, the STH infection continues to pose a major socio-economic challenge and remains an important public health problem in many countries in SSA including Ethiopia [12].

Ethiopia is one of the most populous low-income African countries with one of the highest burdens of STH infections [7]. It is estimated that 79 million people live in areas endemic to STH. This population consists of about 9.1 million pre-school-aged children (pre-SAC), 25.3 million school-aged children (SAC), and 44.6 million adults (above 18 years of age). The number of people living in these areas and requiring treatment for STH is estimated to be 53.6 million people, including 4.6 million pre-SAC, 17.7 million SAC, and 31.3 million adults [13,14]. Low socioeconomic status, poor sanitation, very low latrine coverage and the lack of access to safe drinking water are some of the major factors contributing to an increased risk and high prevalence of STH and other infectious diseases in Ethiopia [14–16].

Morbidity and the burden of the disease resulting from STH infection are directly related to the intensity of infection and its chronic nature [4,17]. Moderate and heavy infection intensity and chronic STH infection can result in and contribute to anemia, malnutrition, growth stunting, low birth weight, physical and cognitive impairment, decreased school performance and hence impacting negatively on economic development [18,19]. Pre-SAC, SAC, women of reproductive age, and adults in high-risk occupations are the vulnerable groups with a significant burden of STH infections [4].

The current global control and prevention strategy for STH infections is based on an integrated approach which includes periodic medicinal treatment (deworming) with single-dose albendazole (400 mg) or mebendazole (500 mg) in the target population and health education on environmental sanitation and personal hygiene [1,4]. These measures are intended to prevent re-infection and reduce soil contamination [1,4,11]. The control efforts have reduced the intensity of infections among pre-SAC and SAC with associated reductions in morbidity in the targeted populations. However, adults and other vulnerable groups in the population are not frequently targeted and remain important reservoirs for reinfection of the treated children as well as others [20].

Ethiopia has set an ambitious national target of eliminating STH as public health problems by 2020, and breaking their transmission by 2025 [21]. Therefore, there is a need to additionally incorporate other groups at risk in the community into the current target population, increasing the impact of the interventions and eliminating STH as a public health concern [22]. Most of the studies conducted in Ethiopia mainly focus on SAC, and data regarding the prevalence and distribution among adults are scarce [23]. Therefore, the present study was undertaken to determine the prevalence and intensity of soil-transmitted helminth infections as well as associated risk factors among the household heads (HH) in the peri-urban areas of Jimma town, Oromia, Ethiopia.

## Materials and methods

### Study design and setting

A community-based cross-sectional study was conducted in five Peri-urban kebeles (the smallest administration unit in Ethiopia, which comprises of a population of approximately 5,000 household populations) of Jimma town from May to July 2021. Jimma town is located 356 km south-west of the capital city, Addis Ababa. Geographically, it is located at a latitude and longitude of 7˚ 40' N, 36˚ 50' E and between altitudes of 1750 and 2000m above sea level. The town is characterized by a semi-arid type of climate with an average annual rainfall of 800–2,500 mm and a temperature range of 20–30˚C [24]. Based on the 2007 Ethiopian central statistical agency census report and current projections, the total population of the town was estimated to be 205,384 in 2018 [25]. The town is divided into 17 kebeles (12 urban and 5 peri-urban kebeles). Jiren, Bore, Horagibe, Kofe, and Ifabula are the names of the five peri-urban kebeles that surround Jimma town and are covered by the study (Fig 1).

### Sample size and sampling procedure

The sample size was determined using a single population proportion formula and assuming a 48.25% prevalence rate of STH derived from a previous study conducted in Jimma town [26]. By using a marginal error of 5%, a 95% confidence interval, and 10% non-response rates, the total sample size was calculated to be 422 household heads. All the five kebeles in the peri-urban areas of Jimma town were included in the study. Using the stratified random sampling method, the sample size for each kebeles (stratum) was determined by proportionate allocation to the total number of households in the kebeles. Finally, the head of the household in each of

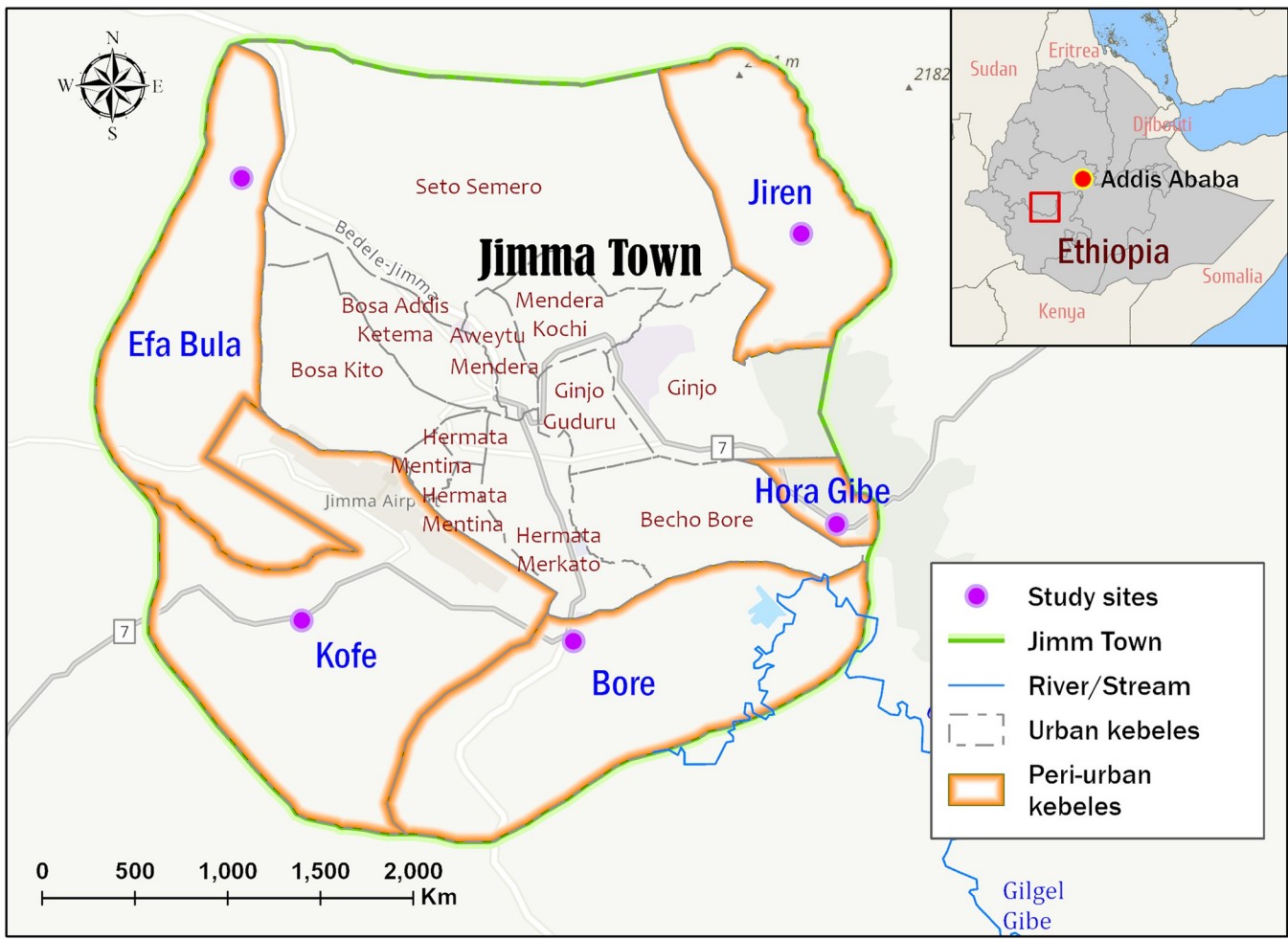

**Fig 1. Location of the study sites: The five peri-urban kebeles in Jimma town, Oromia, Ethiopia.**

the selected households in the five peri-urban kebeles was selected and asked for consent to be included in the study (Fig 2).

## Qualitative data collection and processing

Data on demographic and socioeconomic characteristics, personal and household level sanitation and hygiene practices, as well as other potential risk factors for STH infections, were collected using a pretested semi-structured questionnaire and a checklist prepared for the purpose of this study. The questionnaire was first prepared in English and then translated into the local language (Afan Oromo). Trained community health workers conversant in the local language made house-to house visits and collected the qualitative data from each of the HH heads after obtaining written informed consent.

## Stool sample collection and examination

A sufficient amount of stool specimens (approximately 5 g) was collected from each participant by an experienced laboratory technologist using a clean, leak-proof, screw-cap stool cup labeled with a unique identifier (UID) and following standard operating procedures. All

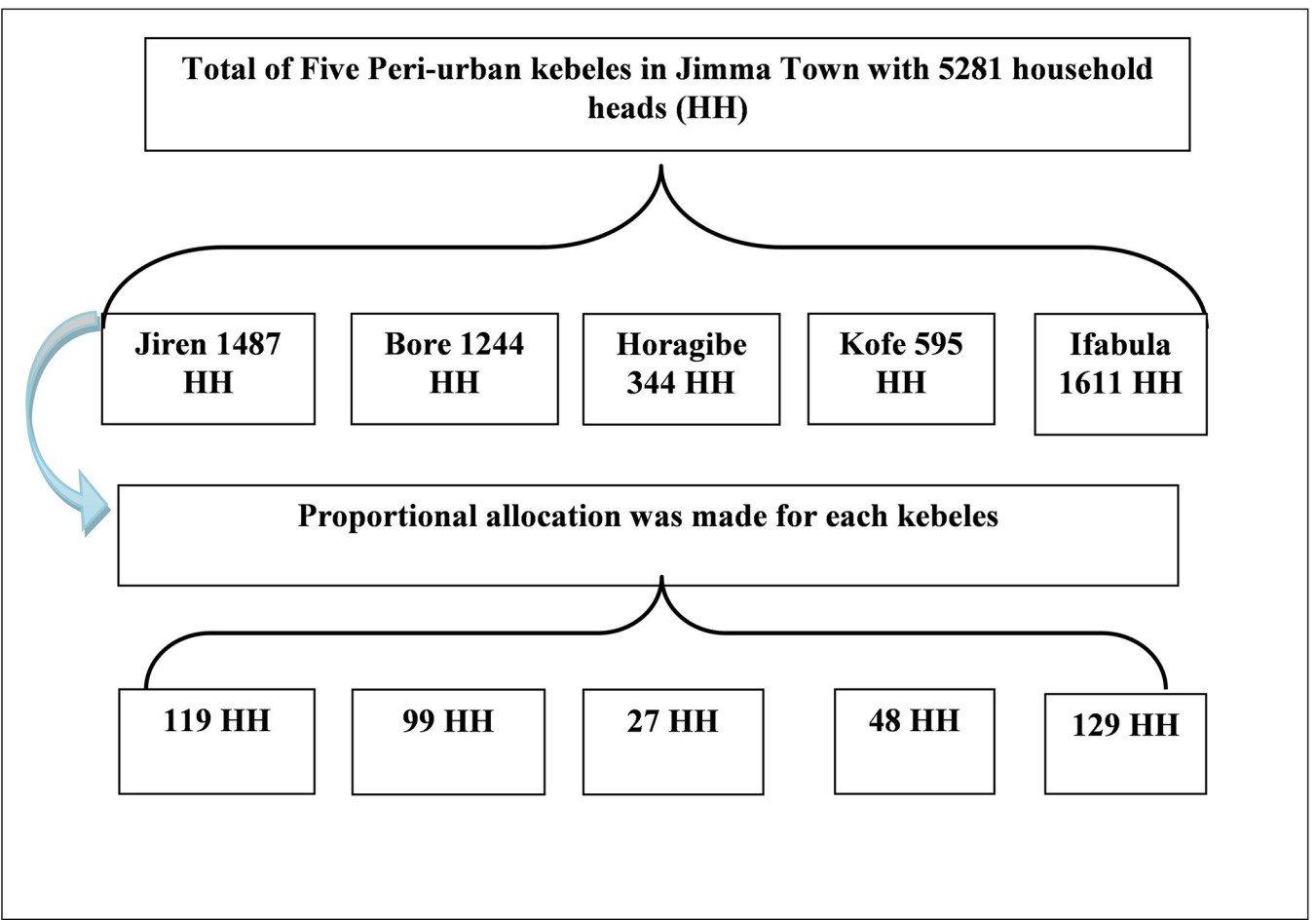

**Fig 2. Diagrammatic representation of the study kebeles (area) and their respective household heads (HH).**

collected specimens were checked for their labels, quantity, and procedure of collection. The stool specimen was then stored in a plastic bag and transported to the medical parasitology laboratory at Jimma University within 2 hours. The stool specimens were processed using a single Kato-Katz smear and examined microscopically to detect ova of the STHs and to quantify the intensity of infections based on fecal egg count (eggs per gram) and according to WHO guidelines [27]. All the Kato-Katz slides were prepared immediately after the arrival of the stool specimens in the laboratory and read between 20 and 30 minutes after slide preparation, which is less than the WHO recommends reading time (30 to 60 minutes) to have better results for hook worm [28].

## Ethics approval and consent to participate

Ethical clearance was obtained from Jimma University Research and Ethics Review Board (IRB) (Reference No. IHRPGD/227/13 in Ethiopian calendar) and permission to conduct the study was obtained from Jimma Zone Health Bureau. Informed written consent was obtained from each household head prior to involvement in the study. Confidentiality of an individual's information was maintained during data collection, analysis, and interpretation. Individuals found positive for any STH were treated with albendazole (400 mg/day, P.O. for three days by the study nurse.

## Data analysis

The collected data was manually checked for completeness and consistency before being coded and double-entered into EpiData version 4.6. The data was then exported to SPSS statistical software version 25.0 for further analysis. Normality was checked for continuous variables using histograms, PP plots and Q-Q plots. The dependent variable was any STH infection among the household heads. The independent variables such as demographic, socioeconomic, environmental and behavioral factors were treated as categorical variables and presented as frequencies and percentages. A chi-square test was performed where appropriate to identify any association between STH infection and the independent variables. Both, bivariate and multivariable logistic regression analyses were employed to identify the candidate variables and potential risk factors for STHs respectively. Odds ratios with 95% confidence intervals were calculated, and a p-value less than 0.05 was considered statistically significant. The wealth index of the household was assessed using the demographic health survey (DHS) [29,30].

## Results

### Socio-demographic characteristics of the household heads

A total of 422 households with an eligible head of household in the peri-urban community were included in this study with a response rate of 89% (n = 376). Of the 376 households that participated in the study, 19.9% and 80.1% of the households were headed by women and men, respectively. The mean age of the household heads was 40.92 ± 9.3 (mean ± SD) years, with the age range from 26 to 75 years. The family size ranged from 2 to 10 individuals per household, with a mean of 4.8 ± 1.5. The majority of the household heads were married (85.4%), had primary school or less education (78.5%), were farmers (32.9%) by occupation, and had low income (36.7%) (Table 1).

### Prevalence and intensity of STH infections

The overall prevalence of STH infection among the household heads in the per-urban Kebele was found to be 18.1%. *A. lumbricoides* was the predominant STH identified with a prevalence rate of 11.4%, followed by *T. trichiura* and Hookworm species with a prevalence of 7.2% and 2.1%, respectively (Table 1).

Most of the STH positive household heads had a single infection (85.3%), while 14.7% of them had multiple infections, which included seven cases of *A. lumbricoides* and *T. trichiura* co-infection (Fig 3). Similarly, most of the study participants infected with *A. lumbricoides* (83.7%), *T. trichiura* (92.6%), and all the eight subjects positive for hookworm eggs had light infection intensity, and only one individual infected with *T. trichiura* had heavy infection intensity (defined by the egg count per gram of stool). The overall geometric mean faecal egg count for *A. lumbricoides*, *T. trichiura*, and hookworms was found to be 607.06 (IQR: 144–2136), 122.59 (IQR: 72–120, and 110.64 (IQR: 72–180), respectively (Table 2).

### Household water supply, sanitation and hygiene conditions

Overall, about 97.1% and 88.6% of the peri-urban households had access to an improved source of water for drinking and domestic use, respectively. Protected well/spring was the most common sources of water in households for both drinking (46.3%) and domestic use (51.6%). Water piped into the household's dwelling was the second most commonly used water source for drinking (35%) and domestic use (19.4%). Most peri-urban households (88%) had private toilet facilities, while 10.1% and 1.9% shared toilet facilities with neighbors and practiced open defecation, respectively. About 56.1% of the households surveyed used

**Table 1. Prevalence of STH and its distribution by socio-demographic characteristics of the household heads in the peri-urban area of Jimma town, Oromia, Ethiopia, May to July 2021 (n = 376).**

| Variables | Categories | Prevalence rate | | | | |
|---|---|---|---|---|---|---|
| | | Any STH | *A. lumbricoides* | *T. trichiura* | Hookworm species | Total |
| | | n (%) | n (%) | n (%) | n (%) | n (%) |
| | Total | 68 (18.1) | 43 (11.4) | 27 (7.2) | 9 (2.4) | 376 (100) |
| Kebeles | Bore | 30(27.8) | 19(17.6) | 15(13.9) | 4(3.7) | 108(28.7) |
| | Kofe | 9(18.0) | 6(12.0) | 2(4.0) | 1(2.0) | 50(13.3) |
| | Hora gibe | 4(12.1) | 2(6.1) | 3(9.1) | 0(0.0) | 33(8.8) |
| | Jiren | 11(10.6) | 9(8.7) | 3(2.9) | 0(0.0) | 104(27.7) |
| | Ifabula | 14(17.3) | 7(8.6) | 4(4.9) | 4(4.9) | 81(21.5) |
| Age | 18–39 | 37 (20.2) | 21 (11.5) | 18 (9.8) | 4 (2.2) | 183 (48.7) |
| | ≥ 40 | 30 (18.1) | 21 (12.7) | 8 (4.8) | 5 (3.0) | 193 (51.3) |
| Sex | Male | 58 (19.3) | 38 (12.6) | 23 (7.6) | 6 (2.0) | 301 (80.1) |
| | Female | 10 (13.3) | 5 (6.7) | 4 (5.3) | 3 (4.0) | 75 (19.9) |
| Family size | <5 | 34 (19.1) | 23 (12.9) | 13 (7.3) | 6 (3.4) | 178 (47.3) |
| | >5 | 34 (17.2) | 20 (10.1) | 14 (7.1) | 3(1.5) | 198 (52.7) |
| Educational status (school years) | Unable to read and write | 18 (32.1) | 14 (25.0) | 4 (7.1) | 4 (7.1) | 56 (14.9) |
| | Primary (1–8) | 39 (16.3) | 23 (9.6) | 20 (8.4) | 3 (1.3) | 239 (63.6) |
| | Secondary (9–12) | 9 (16.7) | 5 (9.3) | 2 (3.7) | 2 (3.7) | 54 (14.4) |
| | High school and above 12 | 2 (7.4) | 1 (3.7) | 1 (3.7) | 0 (0.0) | 27 (7.2) |
| Marital status | Single [1] | 5 (9.1) | 3 (5.5) | 2 (3.6) | 2 (3.6) | 55 (14.6) |
| | Married | 63 (19.6) | 40 (12.5) | 25 (7.8) | 7 (02.2) | 321 (85.4) |
| Occupation | Farmer | 23 (19.0) | 14 (11.6) | 11 (9.1) | 1 (0.8) | 121 (32.2) |
| | Civil servant | 12 (12.1) | 7 (7.1) | 4 (4.0) | 2 (2.0) | 99 (26.3) |
| | Merchant | 8 (10.8) | 6 (8.1) | 5 (6.8) | 1 (1.4) | 74 (19.7) |
| | Daily laborer | 24 (35.8) | 15 (22.4) | 7 (10.4) | 5 (7.5) | 67 (17.8) |
| | Others* | 1 (6.7) | 1 (6.7) | 0 (00) | 0 (0.0) | 15 (4.0) |
| Wealth status | Low | 38 (27.5) | 24 (19.4) | 15 (10.9) | 5 (3.6) | 138 (36.7) |
| | Medium | 15 (14) | 9 (8.4) | 3 (2.8) | 3(2.8) | 107 (28.5) |
| | High | 15 (11.5) | 10 (7.6) | 9 (6.9) | 1 (0.8) | 131 (34.8) |

Others* includes drivers, retired, tailor, unemployed.

Single[1]; includes divorced, separated and, widowed/widower.

improved toilet facilities, but only 11.1% of them had hand washing facilities in their premises. Moreover, 67.6% of the households disposed of solid waste in open fields and 94.7% of them drained the liquid waste directly into the garden (Table 3, S1 and S2 Tables).

### Risk factor analysis for STH infections

Multivariable logistic regression analysis was performed after selecting the candidate variables through bivariate logistic regression analysis. The binary logistic regression revealed an association between sex, marital status, educational status, occupational status, wealth status, having own latrine, habits of hand washing after defecation, habits of hand washing before meals, washing vegetable before consumption, washing/peeling of fruits before eating, hand washing after contact with soil, shoe wearing habits and status of fingernails at p-value ≤ 0.25. After adjusting for potential confounding variables in the multivariable logistic regression model, wealth status, hand washing habits before meals, fingernails status and toilet facility types were found to have a significant association with STH at p-value <0.05 with 95% CI and AOR.

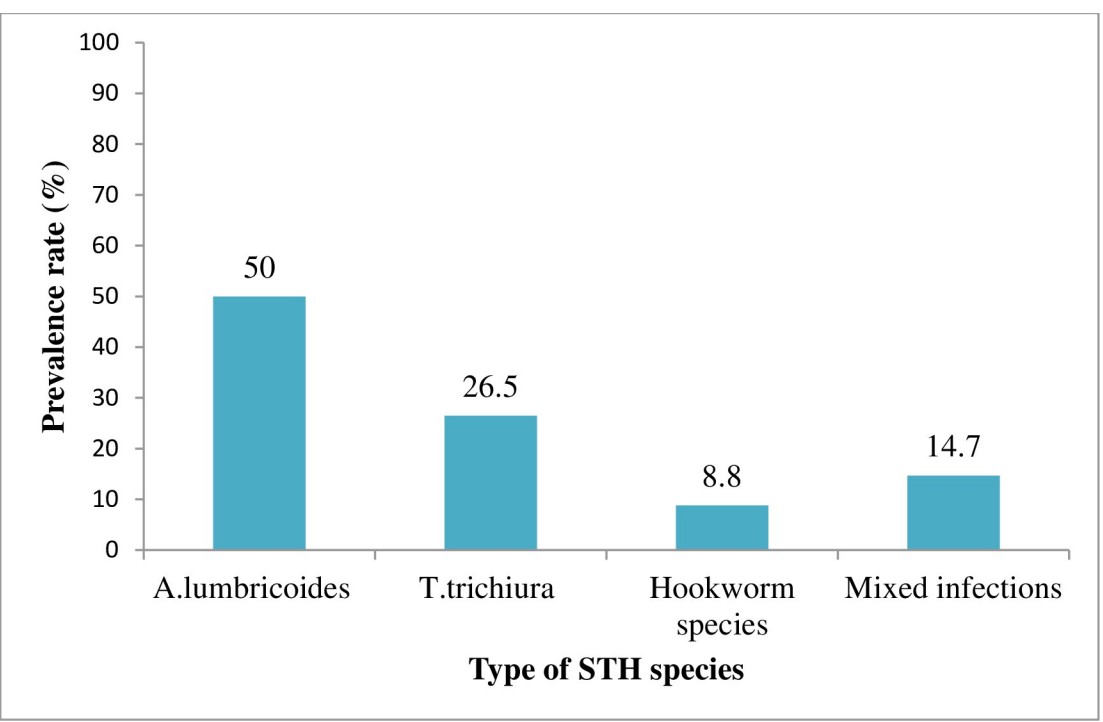

**Fig 3. Prevalence of single and multiple STH species.**

The odds of having STH were was seven times higher among household heads who do not always wash their hands before meals compared to household heads who always wash their hands before meals (AOR = 7.07; 95% CI: 1.79–27.88, p = 0.005). The odds of having STH was 3 times higher among household heads with untrimmed hand finger-nails than among those with trimmed fingernails (AOR = 3; 95% CI: 1.59–5.65, p = 0.001). Household heads who were within the lower wealth percentile were 2.7 times more likely to be infected with STHs as compared to household heads who were wealthier (AOR = 2.7; 95% CI: 1.31–5.50, P = 0.007). Household heads who used unimproved toilet facilities were two times more likely to be infected with STH as compared to household heads who possessed improved toilet facilities (AOR = 2.06; 95% CI: 1.13–3.76, p = 0.017), (Table 4).

**Table 2. Intensity of STH infections in the peri-urban area of Jimma town, Oromia, Ethiopia, May to July 2021 (n = 376).**

| Infection intensity | Soil-transmitted helminths | | |
|---|---|---|---|
| | *A. lumbricoides* | *T. trichiura* | Hookworm species |
| | n (%) | n (%) | n (%) |
| Light | 36 (83.7) | 25 (92.6) | 9 (100) |
| Moderate | 7 (16.3) | 1 (3.7) | 0 |
| Heavy | 0 | 1 (3.7) | 0 |
| Geometric mean (EPG) | 607.06 | 122.59 | 110.64 |
| Total | 43 | 27 | 9 |

*EPG = Eggs per gram of feces. **Infection intensity = *A. lumbricoides* (Light infection 1–4,999 EPG, Moderate infection 5,000–49,999 EPG, and heavy infection >50,000 EPG), *T.trichiura* (Light infection 1–999 EPG, Moderate infection 1,000–9,999 EPG, and heavy infection >10,000 EPG) and *hookworm* (Light infection 1–1,999 EPG, Moderate infection 2,000–3,999 EPG, and heavy infection >4,000 EPG).

**Table 3. Water supply, sanitation and hygiene conditions of the households and distribution of STH in the peri-urban area of Jimma town, Oromia, Ethiopia, May to July 2021 (n = 376).**

| Variables | Categories | STH prevalence | | Total n (%) |
|---|---|---|---|---|
| | | Positive n (%) | Negative n (%) | |
| **Hygiene** | | | | |
| **Hand washing before meal** | Always | 61 (16.7) | 304 (3.3) | 365 (97.1) |
| | often | 7 (63.6) | 4 (36.4) | 11 (2.9) |
| **Hand washing after defecation** | Always | 59 (16.5) | 299 (83.5) | 358 (95.2) |
| | often | 9 (50) | 9 (50) | 18 (4.8) |
| **Washing vegetable before eating** | Always | 36 (15.5) | 192 (84.2) | 228 (60.8) |
| | Often | 19 (17.8) | 88 (82.2) | 107 (28.5) |
| | Sometimes | 13 (32.5) | 27 (67.5) | 40 (10.7) |
| **Washing / peeling fruits before eating** | Always | 29 (14.6) | 170 (85.4) | 119 (52.9) |
| | Often | 27 (20.6) | 104 (79.4) | 131 (34.8) |
| | Sometimes | 12 (26.1) | 34 (73.9) | 46 (12.2) |
| **Status of fingernails** | trimmed | 43 (14) | 265 (86) | 308 (82.1) |
| | untrimmed | 25 (37.3) | 42 (62.7) | 67 (17.9) |
| **Hand washing facility on premises** | Available with soap and water | 2 (11.8) | 15 (88.2) | 17 (4.9) |
| | Available without soap and water | 4 (18.2) | 18 (81.8) | 22 (6.3) |
| | Not available | 54 (17.4) | 257 (82.6) | 311 (88.9) |
| **Sanitation** | | | | |
| **Toilet availability** | Yes | 60 (17.1) | 290 (82.9) | 350 (93.1) |
| | No | 8 (30.8) | 18 (69.2) | 26 (6.9) |
| **Status of the Toilet facilities** | Improved | 23 (10.6) | 188 (89.1) | 211 (56.1) |
| | Unimproved | 45 (27.3) | 120 (72.7) | 165 (43.9) |
| **Shoe wearing habit** | Always | 35 (15.8) | 187 (84.2) | 222 (59.0) |
| | Often | 27 (19.6) | 111 (80.4) | 138 (36.7) |
| | Sometimes | 6 (37.5) | 10 (62.5) | 16 (4.3) |
| **Water source** | | | | |
| **Drinking water source** | Protected/improved | 67 (18.4) | 298 (81.6) | 365 (97.1) |
| | unprotected | 1 (9.1) | 10 (90.9) | 11 (2.9) |
| **Water source for domestic use** | Protected/improved | 63 (18.9) | 270 (81.1) | 333 (88.6) |
| | unprotected | 5 (11.6) | 38 (88.4) | 43 (11.4) |
| **Waste disposal** | | | | |
| **Proper solid waste disposal** | yes | 60 (18.0) | 274 (82.0) | 334 (88.8) |
| | No | 8 (19.0) | 34 (81.0) | 42 (11.2) |
| **Proper liquid waste disposal** | yes | 3 (42.9) | 4 (57.1) | 7 (1.9) |
| | No | 65 (17.6) | 304 (82.4) | 369 (98.1) |

## Discussion

The overall prevalence of any STH infection among the household heads living in the peri-urban area of Jimma town was found to be 18.1%, and most of these positive cases were single infections (85.3%) and light intensity infections (88.5%). *A. lumbricoides* was the predominant STH, followed by *T. trichiura* and hookworm. The vast majority of households had access to an improved source of water for drinking (97.1%) and domestic use (88.8%). Overall, 56.1% of households use improved toilet facilities. In the present study, about 80.1% of households were male-headed while 19.9% of them were female-headed, which is comparable with (78% male

**Table 4. Binary and multivariable logistic regression model to identify factors associated with STH infection among the household heads in the peri-urban area of Jimma town, Oromia, Ethiopia, May to July 2021(n = 376).**

| Variables | Categories | Soil-transmitted helminth | | COR (95% CI) | P-value | AOR (95% CI) | P-value |
|---|---|---|---|---|---|---|---|
| | | Positive n (%) | Negative n (%) | | | | |
| Kebeles | Bore | 30(27.8) | 78(72.2) | 1.84 (0.90–3.76) | 0.094* | 1.22(0.53–2.78) | 0.643 |
| | Kofe | 9(18.0) | 41(82.0) | 1.05(0.42–2.64) | 0.917 | 0.58(0.19–1.62) | 0.284 |
| | Hora gibe | 4(12.1) | 29(87.9) | 0.66(0.20–2.18) | 0.495 | 0.63(0.17–2.21) | 0.470 |
| | Jiren | 11(10.6) | 93(89.4) | 0.56(0.24–1.32) | 0.189* | 0.58(0.23–1.50) | 0.257 |
| | Ifabula | 14(17.3) | 67(82.7) | 1 | | | |
| **Sex** | Male | 58(19.3) | 243(80.7) | 1.55(0.75–3.20) | 0.235* | 1.37(0.60–3.09) | 0.447 |
| | Female | 10(13.3) | 65(86.7) | 1 | | 1 | |
| **Marital status** | Single[1] | 5(9.1) | 50(90.9) | 0.410(0.15–1.06) | 0.068* | 0.38(0.14–1.04) | 0.060 |
| | Married | 63(19.6) | 258(80.4) | 1 | | 1 | |
| **Educational status** | Unable to read and write | 18(32.1) | 38(67.9) | 5.92(1.26–27.77) | 0.024* | 1.60(0.29–8.73) | 0.586 |
| | Primary (1–8) | 39(16.3) | 200(83.7) | 2.43(0.55–10.71) | 0.238* | 0.96(0.19–4.63) | 0.959 |
| | Secondary (9–12) | 9(16.7) | 45(83.3) | 2.50(0.50–12.49) | 0.264 | 1.83(0.34–9.83) | 0.478 |
| | High school & above | 2(7.4) | 25(92.6) | 1 | | 1 | |
| **Occupation** | Farmer | 23(19.0) | 98(81.0) | 3.28(0.41–26.27) | 0.262 | 2.17(0.25–18.45) | 0.477 |
| | Civil servant | 12(12.1) | 87(87.9) | 1.93(0.23–16.03) | 0.542 | 1.45(0.16–12.76) | 0.736 |
| | Merchant | 8(10.8) | 66(89.2) | 1.69(0.19–14.67) | 0.631 | 1.14(0.12–10.47) | 0.907 |
| | Daily laborer | 24(35.8) | 43(64.2) | 7.81(0.96–63.13) | 0.054* | 4.58(0.53–39.30) | 0.167 |
| | Others | 1(6.7) | 14(93.3) | 1 | | 1 | |
| **Wealth status** | Low | 38(27.5) | 100(72.5) | 2.94(1.53–5.66) | 0.001* | 2.7(1.31–5.50) | **0.007**** |
| | Medium | 15(14.0) | 92(86.0) | 1.26(0.59–2.71) | 0.553 | 1.53(0.67–3.48) | 0.303 |
| | High | 15(11.5) | 116(88.5) | 1 | | 1 | |
| **Latrine availability** | Yes | 60(17.1) | 290(82.9) | 1 | | 1 | |
| | No | 8(30.8) | 18(69.2) | 2.15(0.89–5.17) | 0.088* | 0.58(0.19–1.75) | 0.335 |
| **Toilet facility** | Improved | 23(10.6) | 188(89.1) | 1 | | 1 | |
| | unimproved | 45(27.3) | 120(72.7) | 3.06(1.76–5.32) | <0.001* | 2.06(1.13–3.76) | **0.017**** |
| **Source of Water for domestic use** | protected | 63(18.9) | 270(81.1) | 1 | | 1 | |
| | Unprotected | 5(11.6) | 38(88.4) | 0.56(0.21–1.49) | 0.248* | 0.49(0.17–1.44) | 0.194 |
| **Hand washing after defecation** | Always | 59(16.5) | 299(83.5) | 1 | | 1 | |
| | Often | 9(50) | 9(50) | 5.07(1.9–13.3) | 0.001* | 1.60(0.44–5.77) | 0.468 |
| **Hand washing before meals** | Always | 61(16.7) | 304(3.3) | 1 | | 1 | |
| | Often | 7(63.6) | 4(36.4) | 8.72(2.48–30.71) | 0.001* | 7.07(1.79–27.88) | **0.005**** |
| **Washing vegetable before eating** | Always | 36(15.5) | 192(84.2) | 1 | | 1 | |
| | Often | 19(17.8) | 88(82.2) | 1.15(0.62–2.12) | 0.651 | 1.51(0.45–5.06) | 0.503 |
| | Sometimes | 13(32.5) | 27(67.5) | 2.59(1.21–5.44) | 0.014* | 2.33(0.56–9.61) | 0.239 |
| **Washing/ peeling fruits before eating** | Always | 29(14.6) | 170(85.4) | 1 | | 1 | |
| | Often | 27(20.6) | 104(79.4) | 1.52(0.85–2.71) | 0.155* | 1.42(0.74–2.68) | 0.285 |
| | Sometimes | 12(26.1) | 34(73.9) | 2.07(0.96–4.45) | 0.063* | 0.96(0.39–2.34) | 0.925 |
| **Hand washing after contact with soil** | Always | 54(16.6) | 271(83.4) | 1 | | 1 | |
| | Often | 9(33.3) | 18(66.7) | 2.51(1.07–5.88) | 0.034* | 1.52(0.50–4.64) | 0.458 |
| | Sometimes | 5(20.8) | 19(79.2) | 1.32(0.47–3.69) | 0.596 | 0.97(0.27–3.44) | 0.963 |
| **Shoe wearing habit** | Always | 35(15.8) | 187(84.2) | 1 | | 1 | |
| | Often | 27(19.6) | 111(80.4) | 1.30(0.75–3.26) | 0.354 | 0.79(0.36–1.70) | 0.551 |
| | Sometimes | 6(37.5) | 10(62.5) | 3.21(1.09–9.39) | 0.034* | 1.68(0.43–6.57) | 0.454 |

(*Continued*)

**Table 4.** (Continued)

| Variables | Categories | Soil-transmitted helminth | | COR (95% CI) | P-value | AOR (95% CI) | P-value |
|---|---|---|---|---|---|---|---|
| | | Positive n (%) | Negative n (%) | | | | |
| **Status of fingernails** | trimmed | 43(14) | 265(86) | 1 | | 1 | |
| | untrimmed | 25(37.3) | 42(62.7) | 3.67(2.03–6.62) | <0.001* | 3(1.59–5.65) | **0.001**** |

Key

* = candidate variables at p ≤0.25 in bivariate logistic regression

** predictor variables in multivariate logistic regression at p <0.05.

and 22% female) the Ethiopian national demographic and health survey of 2019 [30]. In general, the wealth status of the household, habits of hand washing before meals, the status of fingernails, and types of toilet facilities showed a statistically significant association with the presence of soil- transmitted helminth eggs in the stool sample.

The results of this study revealed that the STH is prevalent among the adult population and endemic in the study area, despite many years of school-based deworming programs in the town. This implies that if the infected adult populations of the community are left untreated, they may serve as an important reservoir and source of re-infection for the treated children, contributing to the sustained transmission of the STH in the community. Previous studies conducted in various countries have shown that STH reinfections occur rapidly after treatment, particularly for *A. lumbricoides* and *T. trichiura*, and hence have a significant impact on the success of preventive mass chemotherapy [31–35].

In Ethiopia, the number of people living in STH endemic areas is estimated at 79 million, including 9.1 million pre-school-aged children, 25.3 million school-aged children, and 44.6 million adults [13]. In the past decades, Ethiopia has demonstrated a marked and sustained decrease in the prevalence and intensity of STH infections through preventive mass chemotherapy in the targeted populations [21]. However, the levels of infection among older individuals and adults continued to be high, which could pose a significant challenge to the parasite reduction achieved in children and to Ethiopia's national target of eliminating STH as a public health problem by 2020, which wasn't achieved, and breaking their transmission by 2025 in general [7,21].

The result of this study illustrates the importance of addressing the community-based reservoir to reduce the reinfections of the treated children and other at-risk populations through expanding the target population, integrated with health and hygiene education and the provision of adequate sanitation facilities [31,32,34]. A study conducted among agrarian communities of Kogi State, Nigeria recommends the inclusion of all age groups in preventive chemotherapy along with health education and provision of basic sanitation facilities to eliminate STH and break their transmission by 2025 [36,37].

Most of the previous studies conducted on STH in Ethiopia focused on SAC, which is possible because of the Ethiopian priority target population set in the national master plan of neglected tropical diseases (NTDs) control program [13,21,38]. There is a lack of data on the prevalence and burden of STH among pre-SAC and adults, and no studies have been conducted specifically among Ethiopian heads of households [21]. In Ethiopia, the head of the household is generally considered as a reference person and major decision maker in the household unit who controls the household finances and other assets, which may lead to deference in health care access and service utilization among the household members [16,39]. On the other hand, the infection of the household head with intestinal helminths could have an indirect impact on the productivity and wage-earning capacity of the household. A review

conducted by Helen Guyatt indicated that productivity during adulthood could be affected by current infection and associated morbidity, and early infection during childhood [40].

This study is the first to provide information on the prevalence and intensity of STH infection and the associated risk factors among the household heads in the peri-urban communities of Jimma town. Variation in prevalence was seen among the five kebeles, ranging from 10.6% (Hora gibe kebele) to 27.8% (Bore kebele) and the prevalence of any STH were not, however, shown to be statistically associated (p > 0.05). The current study's observed total prevalence of STH infections was 18.1%, which is comparable to previous studies conducted among adult populations in the coast of Kenya (20.7%) [34], Southern Thailand (15.7%) [41], Eastern Côte d'Ivoire (19.5%) [42] and Northwest Ethiopia (20.9%) [43]. However, it was lower than the finding of the studies conducted in the rural community of the southwest Ethiopia (70.3%) [44], Kogi state, Nigeria (45.1%) [36], Cameron's Western region of (26.4%) [45], Ghana's middle-belt (45%) [46] and Guinea Bissau's Bijagos Islands (40%) [47]. Prevalence of STH infections ranging from 3.3 to 51.5% were also reported in the study conducted among the adult population in five communities in Nepal [48]. On the other hand, the prevalence rate in the current study is higher than the finding of the study conducted in central Kenya (0.2%) [49], and in the district of Come in Benin (5.3%) [50].

Regarding the prevalence of STH species, *A. lumbricoides* was the most common STH in the present study (11.4%), followed by *T. trichiura* (7.2%) and hookworm (2.1%). However, *T. trichiura* (66.6%) and hookworm species (19.1%) were found to be the predominant STH species in the study conducted among the rural community of southwest Ethiopia [44] and on the coast of Kenya [34]. In the current study, most of the household heads positive for STH had single infections (85.3%), while only 14.7% of them had multiple infections, which was lower than in the study conducted in rural communities of southwest Ethiopia [44] and the western region of Cameroon [51], with multiple infection rates of 44.2% and 26.4%, respectively. Moreover, most of the study participants infected with STH in the current study had light infection intensity (88.5%), which was comparable with the findings of the study conducted in a rural community in southwest Ethiopia, and among elderly people in rural areas of the southern part of Thailand [41,44]. People with light intensity infections may not usually seek treatment because of not having observable symptoms, which may contribute to environmental contamination and the sustained transmission of the parasites in the community.

In general, the variation in prevalence rate, intensity, and distribution of these STH species among the different communities and populations of the study might be due to differences in the socio-demographics and socioeconomic status of the households, macro-and micro-environmental factors, and host-specific and individual factors which may affect the risk of acquiring or harboring heavy intensity STH infections [52].

In this study, about 97.1% of the peri-urban households had access to an improved source of water for drinking, which is higher than the national figure of 87% for urban households and 61% for rural households [30], Protected well/spring (46.3%) and water piped into the household's dwelling (35%) were found to be the most common sources of water for drinking in the peri-urban households in Jimma town. Our findings are comparable with the findings of the Ethiopia demographic and health survey (2019) [30], which reported that water piped into the household's dwelling, yard, or plot (40%) and water piped into a public tap/standpipe (30%) in the urban households and public taps/standpipes (31%) and protected springs (13%) in the rural households were reported to be the most common sources of water for drinking use.

Overall, 56.1% of the peri-urban households of Jimma town use improved toilet facilities. However, this is in contrast to the finding of the Ethiopia Demographic and Health Survey (2019) [30], which reported that only 20% of Ethiopian households use improved toilet facilities (42% in urban areas and 10% in rural). This variation can be explained by a variety of

factors, including but not limited to differences in the socioeconomic characteristics of the household population, water, sanitation, and hygiene (WASH) facilities and usage.

Low socioeconomic status and poor hygienic behaviors, which include untrimmed hand fingernails and not always washing hands before meals, were found to be significantly associated with STH infections. The odds of having STHs was seven times higher among household heads that do not always wash their hands before meals compared to those household heads that always washed their hands before meals. A similar finding was reported in the study conducted in Bibugn district, northwest Ethiopia [43], and the western region of Cameroon, which indicated that households not washing their hands before meals were more affected by STHs than their counterparts [45].

The odds of having STH were 3 times higher among household heads that had untrimmed hand fingernails compared to household heads with trimmed fingernails. This is in agreement with the study conducted in southwest Ethiopia [53]. The present study also found that household heads in the lowest wealth percentile were 2.7 times more likely to be infected with STHs as compared to household heads in the highest wealth percentile. A similar result was reported from Kenya, Nigeria, Benin, Malaysia, and Indonesia, where individuals with low income were more affected by STHs than their counterparts [34,36,50,54,55]. Here we also found that household heads who possessed unimproved toilet facilities were two times more likely to be infected with STHs as compared to household heads who possessed improved toilet facilities. This has also been described in the study conducted in western Cameron and southern India [51,56].

## Limitation

The major limitation of this study is that the prevalence and infection intensity of STHs were determined by the examination of a single stool specimen from each study participant only. This might not be enough to accommodate the day-to-day and inter-stool variation of egg output. Furthermore, a single Kato-Katz template was examined for each of the stool specimens that might affect the accuracy of the egg count. Both limitations are expected to underestimate the real STH burden in the population investigated, especially as those subjects with low egg counts are missed more easily by the investigation of only one sample.

## Conclusion

The results of this study revealed a nearly moderate level of STH prevalence among household heads in the peri-urban area, which could be a significant reservoir for reinfection of the treated children and other at-risk groups of the community, posing serious challenges to the national targets of eliminating STH as a public health problem. The wealth status of the household, habits of hand washing before meals, and the status of fingernails showed significant associations with the detection of soil-transmitted helminth infection, suggesting a need for prompting health education and improving the socioeconomic status of the community. Moreover, the findings of this study indicate the need for expanding the deworming program to other at-risk groups, or the whole community. The use of only one stool sample might also lead to a certain under-detection of cases with low infection intensity, so the potential reservoir of STH infections in adults might be even greater, strengthening the evidence for more broad scale deworming.

## Supporting information

**S1 Table. Socio-demographic characteristics, sanitation and hygiene practice of the household heads in peri-urban Kebeles in Jimma town.**
(DOCX)

**S2 Table. Prevalence of STH and its distribution by socio-demographic characteristics, sanitation and hygiene practice of the household heads in peri-urban Kebeles in Jimma town.**
(DOCX)

## Acknowledgments

The authors are so grateful to the household heads for their cooperation in participating and providing the necessary information and stool samples. The authors are also thankful to the laboratory technologist in the Parasitology and Microbiology laboratory for collecting and examining the stool samples. The authors are grateful to the department of medical laboratory science, Jimma University for providing laboratory facilities and material support. We would also like to extend our heartiest appreciation to the Jimma town health office for their cooperation. Dr. Ming-Chieh Lee mapped the study sites, for which we are grateful. Special acknowledgments also goes to the Center for International Health at Ludwig-Maximilians-University (CIH-LMU) for facilitation and support of this project.

## Author Contributions

**Conceptualization:** Ahmed Zeynudin, Million Tesfaye.

**Data curation:** Ahmed Zeynudin, Million Tesfaye, Zuber Hajikelil.

**Formal analysis:** Ahmed Zeynudin, Million Tesfaye, Zuber Hajikelil, Andreas Wieser.

**Funding acquisition:** Ahmed Zeynudin, Million Tesfaye, Andreas Wieser.

**Investigation:** Ahmed Zeynudin, Sultan Suleman, Zuber Hajikelil, Abdusemed Husen, Jafer Yasin.

**Methodology:** Ahmed Zeynudin, Million Tesfaye, Khalide Azam, Andreas Wieser.

**Project administration:** Ahmed Zeynudin, Million Tesfaye, Sultan Suleman.

**Resources:** Ahmed Zeynudin, Teshome Degefa, Sultan Suleman, Andreas Wieser.

**Software:** Ahmed Zeynudin.

**Supervision:** Ahmed Zeynudin, Teshome Degefa, Elias Ali Yesuf, Zuber Hajikelil, Abdusemed Husen, Jafer Yasin.

**Validation:** Ahmed Zeynudin, Million Tesfaye, Sultan Suleman, Zuber Hajikelil, Khalide Azam, Abdusemed Husen, Jafer Yasin, Andreas Wieser.

**Visualization:** Ahmed Zeynudin, Million Tesfaye, Zuber Hajikelil, Andreas Wieser.

**Writing – original draft:** Ahmed Zeynudin.

**Writing – review & editing:** Ahmed Zeynudin, Teshome Degefa, Sultan Suleman, Elias Ali Yesuf, Solomon Ali, Khalide Azam, Abdusemed Husen, Jafer Yasin, Andreas Wieser.

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
