## [Decision Letter · Decision Letter 0]

22 Jun 2022

PONE-D-22-15060Prevalence and intensity of soil-transmitted helminth infections and associated risk factors among household heads living in the peri-urban areas of Jimma town, Oromia, EthiopiaPLOS ONE

Dear Dr. Zeynudin,

Thank you for submitting your manuscript to PLOS ONE. After careful consideration, we feel that it has merit but does not fully meet PLOS ONE’s publication criteria as it currently stands. Therefore, we invite you to submit a revised version of the manuscript that addresses the points raised during the review process.

Please find the comments provided by the editor and reviewers below this email. The comments should be adequately addressed in a revised manuscript.

 Please submit your revised manuscript by Aug 06 2022 11:59PM. If you will need more time than this to complete your revisions, please reply to this message or contact the journal office at plosone@plos.org. Please include the following items when submitting your revised manuscript:A rebuttal letter that responds to each point raised by the academic editor and reviewer(s). You should upload this letter as a separate file labeled 'Response to Reviewers'.A marked-up copy of your manuscript that highlights changes made to the original version. You should upload this as a separate file labeled 'Revised Manuscript with Track Changes'.An unmarked version of your revised paper without tracked changes. You should upload this as a separate file labeled 'Manuscript'.If applicable, we recommend that you deposit your laboratory protocols in protocols.io to enhance the reproducibility of your results. Protocols.io assigns your protocol its own identifier (DOI) so that it can be cited independently in the future. For instructions see: https://journals.plos.org/plosone/s/submission-guidelines#loc-laboratory-protocols. Additionally, PLOS ONE offers an option for publishing peer-reviewed Lab Protocol articles, which describe protocols hosted on protocols.io. Read more information on sharing protocols at https://plos.org/protocols?utm_medium=editorial-email&utm_source=authorletters&utm_campaign=protocols.

We look forward to receiving your revised manuscript.

Kind regards,

Hesham

Hesham M. Al-Mekhlafi, PhD

Academic Editor

PLOS ONE

Journal Requirements:

Additional Editor Comments:

The manuscript needs a more careful proofread by a native speaker to correct some errors.

Please, prepare your revised manuscript following journal's style and format.

Reviewers' comments:

Reviewer's Responses to Questions

**Comments to the Author**

1. Is the manuscript technically sound, and do the data support the conclusions?

Reviewer #1: Yes

Reviewer #2: Yes

2. Has the statistical analysis been performed appropriately and rigorously? 

Reviewer #1: Yes

Reviewer #2: Yes

3. Have the authors made all data underlying the findings in their manuscript fully available?

Reviewer #1: Yes

Reviewer #2: Yes

4. Is the manuscript presented in an intelligible fashion and written in standard English?

Reviewer #1: Yes

Reviewer #2: Yes

5. Review Comments to the Author

Reviewer #1: Manuscript Title:

Prevalence and intensity of soil-transmitted helminth infections and associated risk

factors among household heads living in the peri-urban areas of Jimma town, Oromia, Ethiopia

Reviewer’s decision: Accept with Minor Corrections

The manuscript is of high quality considering the amount of work-done and analysis made. It is also well written. The introduction and rationale for the study is sound. The methodology section is also very well described, without scientific jargon. The result section is also well written and tables well presented. The authors claimed that the prevalence of 18.1% is low, it would be nice to see the prevalence across each communities studied to have a better insight. In the supplementary file 2, one of the communities has a prevalence as high as 27.8%. It would be a great point to talk about these dynamics in the manuscript abstract and discussion.

Reviewer’s Comment

TOPIC:

The title of this manuscript is appropriate and concise,

ABSTRACT:

This is a well written abstract. However authors should confirm the word limit for the abstract section.

INTRODUCTION

Line 68: recast as..”…lack of access to ...

Line 74-75: recast as “hence impacting negatively on economic development….

Line 79-80? Please mention the medicine used during the treatment,

Line 80: Remove the words “to eliminate infecting worms”

Lin 82-83: Please provide a reference here

Line 85: replace “as well as others” with “ and other vulnerable groups”

Materials and Methods

Line 144” mention the specific name of the medicine and the dosage e.g. Albendazole 40mg/kg or so

RESULTS

Line 163: Maintain a 1-digit decimal point. So, 9.3 instead of 9.285

Table showing the prevalence by communities is missing. It would be nice to see how the overall 18.1% STH prev is distributed across the communities studied.

One great suggestion that could help is the rearrangement of the tables, the table on intensity can follow directly after table 1.

Table 3: Authors should provide a footnote explain the infection intensity, and the acronym EPG

Line 196-198: Since it wasn’t reported in the manuscript table, how useful are these lines here.

DISCUSSION

Line 269: needs a reference

Line 272: Rather than using the word failed, authors could recast that the target was not met

Line 353: remove the comma after from,

Line 353: replace the semi-colon with “with”

Reviewer #2: A community-based cross sectional study by Ahmed Zeynudin and others was conducted in Ethiopia from May to July 2021. The study targeted household populations (18 years and above) and aimed at determining the prevalence of STH and risk factors in Jiren, Bore, Horagibe, Kofe and Ifabula Peri-urban kebeles surrounding the Jimma town. While the study is not novel, it provided an update on the STH prevalence and the associated risk factors.

Here are some comments that I hope are constructive for manuscript improvement:

I: General comments

1. It will be useful if the author(s) can provide a country map showing the location of the five selected peri-urban kebeles of study.

2. The prevalence of STH in this study was based on the Kato-Katz microscopy, however, hookworms, in particular, was expected to be underestimated using this technique especially when light infections are common.

3. Please include more details on the Kato-Katz smear preparation and microscopic examination, specimens processing and examination timing, as there is no reference provided in the methodology.

4. Table 1 showed the prevalence of STH in different variable subgroups. The age group >60 years consisted of 27 households which are very low compared to other subgroups, and only one individual was found infected with Ascaris lumbricoides and Trichuris trichiura parasites. I recommend merging this sub-group into the previous one.

5. In table 2, no subjects under “Sometimes” for the washing hand hygiene. Better to delete them from the table.

6. Line 184-185: households who drained the liquids directly into the garden = 94.9%. It should be 94.7% (according to S2 file).

7. Single infections were 83.8% in the abstract (line 32) and discussion (line 247) while 85.1% in the results (line 199) and 83.1% in line 311 in the discussion. Please check and correct.

8. Line 192-195: no need for the CI to present the prevalence (%) of parasite detection.

II: Editing issues:

• Abstract line 16: soil transmitted helminthes: change to “soil-transmitted helminths” (correct elsewhere in lines 41, 44).

• Line 22: A community based: change to “A community-based” (correct elsewhere in line 97).

• Line 23: May to July, 2021: delete comma (similarly in line 99).

• Line 30: A. lumbricoides: italic.

• Line 31: T. trichiura: italic.

• Line 38: still remain: delete still (redundant).

• Line 60: major: change to “a major”.

• Line 68: socio- economic: remove space.

• Line 68: lack of accesses: change to “access”.

• Line 71: is: “are”.

• Line 98: population: change to “a population”.

• Line 102: temperature: change to “a temperature”.

• Line 129: Sufficient: change to “A sufficient”.

• Line 129: were collected: change to “was”.

• Line 130: leak proof: change to “leak-proof”.

• Line 131: unique Identifier (UID): change to “unique identifier” or “Unique Identifier” (UID).

• Line 133: specimen were: change to “specimen was”.

• Line 148: sold waste: change to “solid”.

• Line 156: was assess: change to “ assessed”.

• Line 182: About 56.1% the households: change to “of” the households.

• Line 196-197: S. mansoni, H. nana and E. vermicularis: full genera names.

• Line 199: had single infection: change to “a single”.

• Line 220: washing / peeling: remove spaces.

• Line 222: p- value: remove space.

• Line 222: finger nail: change to fingernails (correct elsewhere in 224, 230, 231, 253, 374 and in tables 2 and table 4).

• Line 250: about 80.1% households: “of” households.

• Line 272: as public health problem: “a public”.

• Line 289: indirect: “an indirect”.

• Line 336: finding the Ethiopia Demographic: “of” the Ethiopia Demographic.

• Line 347: counter parts: one word.

• Line 374: soil- transmitted: remove space.

• Table 1: N (%): change to “n” (%) (correct elsewhere for consistency).

• Table 3 and Table 4: No (%): change to “n” (%).

• Line 247: (88,5%): (88.5%).

• Line 308: (19.1%). were found: remove the full stop after the bracket.

• Line 309: Species: “species” (uncapitalize the word).

• Line 311: (83.1 %,): remove the comma after correcting the number (see comment 6).

• The AOR for untrimmed hand fingernail was 2.99, change the 2.9 (line 229) to 3. Similarly, wealth status (2.7) in line 232.

III: References:

Many references are not according to the journal style, please correct accordingly.

6. PLOS authors have the option to publish the peer review history of their article (what does this mean?). If published, this will include your full peer review and any attached files.

Reviewer #1: No

Reviewer #2: **Yes: **Wahib M. Atroosh

---

## [Author Response · Author response to Decision Letter 0]

27 Jul 2022

Reviewer #1: Manuscript Title:

Prevalence and intensity of soil-transmitted helminth infections and associated risk

factors among household heads living in the peri-urban areas of Jimma town, Oromia, Ethiopia

Reviewer’s decision: Accept with Minor Corrections

The manuscript is of high quality considering the amount of work-done and analysis made. It is also well written. The introduction and rationale for the study is sound. The methodology section is also very well described, without scientific jargon. The result section is also well written and tables well presented. The authors claimed that the prevalence of 18.1% is low, it would be nice to see the prevalence across each communities studied to have a better insight. In the supplementary file 2, one of the communities has prevalence as high as 27.8%. It would be a great point to talk about these dynamics in the manuscript abstract and discussion.

RESPONSE:

• “Variation in prevalence was seen among the five kebeles, ranging from 10.6% (Hora gibe kebele) to 27.8% (Bore kebele) and the prevalence of any STH were not, however, shown to be statistically associated (p > 0.05).”

Reviewer #1

Reviewer’s Comment

1. TOPIC:

The title of this manuscript is appropriate and concise,

2. ABSTRACT:

This is a well written abstract. However authors should confirm the word limit for the abstract section.

RESPONSE: 

• The abstract is corrected to 300 word limit as per the comments and guideline

3. INTRODUCTION

Line 68: recast as..”…lack of access to .

• Corrected - line 71

Line 74-75: recast as “hence impacting negatively on economic development….

• Corrected - line 77

Line 79-80? Please mention the medicine used during the treatment,

• Corrected (medicine used during the treatment is mentioned) - line 81-82

Line 80: Remove the words “to eliminate infecting worms”

• Removed - line 82 

 Lin 82-83: Please provide a reference here

• Reference is provided for line 82-83 (reference No: 1 & 4)

 Line 85: replace “as well as others” with “ and other vulnerable groups”

• Corrected - line 87

4. MATERIALS AND METHODS

Line 144” mention the specific name of the medicine and the dosage e.g. Albendazole 40mg/kg or so

RESPONSE: 

• Specific name and dosage of the drugs are incorporated - line 154

• Albendazole (400 mg/day, P.O. for three days )

5. RESULTS

Line 163: Maintain a 1-digit decimal point. So, 9.3 instead of 9.285

• Corrected - line 174

Table showing the prevalence by communities is missing. It would be nice to see how the overall 18.1% STH prev is distributed across the communities studied.

RESPONSE:

• Data showing the distribution of the STH prevalence across the different communes are added to 

o Table 1(- line 185-190) & 

o Table 4(- line 281-285)

One great suggestion that could help is the rearrangement of the tables, the table on intensity can follow directly after table 1.

RESPONSE:

• Tables are rearranged

o Table 3 rearranged and relabeled as Table 2(- line 207) following table 1 and paragraph following the table - line 192-199

o Table 2 rearranged and relabeled as table 3) - line 229

Table 3 (now table 2): Authors should provide a footnote that explains the infection intensity, and the acronym EPG

• RESPONSE: 

o Footnote and acronym added as follows - line 210-213

o *EPG = Eggs per gram of feces. **Infection intensity = A. lumbricoides (Light infection 1–4,999 EPG, Moderate infection 5,000-49,999 EPG, and heavy infection >50,000 EPG), T.trichiura (Light infection 1-999 EPG, Moderate infection 1,000-9,999 EPG, and heavy infection >10,000 EPG) and hookworm (Light infection 1-1,999 EPG, Moderate infection 2,000-3,999 EPG, and heavy infection >4,000 EPG)

Line 196-198: Since it wasn’t reported in the manuscript table, how useful are these lines here.

• Removed from the manuscript - line 184

6. DISCUSSION

Line 269: needs a reference

o Reference inserted - line 312 (reference no 21)

Line 272: Rather than using the word failed, authors could recast that the target was not met

o Corrected as “which wasn't achieved.” - line 315

Line 353: remove the comma after from,

• comma removed - line 403

Line 353: replace the semi-colon with “with”

• Corrected as “A similar result was reported from Kenya, Nigeria, Benin, Malaysia, and Indonesia, where individuals with low income were more affected by STHs than their counterparts. - line 402-404

Reviewer #2: 

A community-based cross sectional study by Ahmed Zeynudin and others was conducted in Ethiopia from May to July 2021. The study targeted household populations (18 years and above) and aimed at determining the prevalence of STH and risk factors in Jiren, Bore, Horagibe, Kofe and Ifabula Peri-urban kebeles surrounding the Jimma town. While the study is not novel, it provided an update on the STH prevalence and the associated risk factors.

Here are some comments that I hope are constructive for manuscript improvement:

I: GENERAL COMMENTS

1. It will be useful if the author(s) can provide a country map showing the location of the five selected peri-urban kebeles of study.

• RESPONSE: - line 111

o country map showing the location of the five selected peri-urban kebele of study is added to the main manuscript as 

o Fig 1. Location of the study sites: the five peri-urban kebeles surrounding Jimma town, Oromia, Ethiopia

2. The prevalence of STH in this study was based on the Kato-Katz microscopy, however, hookworms, in particular, was expected to be underestimated using this technique especially when light infections are common.

3. Please include more details on the Kato-Katz smear preparation and microscopic examination, specimens processing and examination timing, as there is no reference provided in the methodology.

RESPONSE for comment No 2 & 3

The following sentences were added to the main manuscript - line 144-147

o All the Kato-Katz slides were prepared immediately after the arrival of the stool specimens in the laboratory and read between 20 and 30 minutes after slide preparation, which is less than the WHO recommends reading time (30 to 60 minutes) to have better results for hook worm

• Reference 

o Additionally, New reference No= 27 and 28 explaining the detailed procedure recommended by the WHO was added to explain the concern of the reviewer = line 524 and 526

4. Table 1 showed the prevalence of STH in different variable subgroups. The age group >60 years consisted of 27 households which are very low compared to other subgroups, and only one individual was found infected with Ascaris lumbricoides and Trichuris trichiura parasites. I recommend merging this sub-group into the previous one.

• RESPONSE: 

o The age category are merged as per the reviewer’s comments and as follows = line 185 ( table 1) and line 281(table 4)

18-39

≥ 40 

o Consecutively (as indicted in the truck changes) 

prevalence figures for the new age category in table 1 is changed accordingly 

AOR (95% CI) and P-value in table 4 are changed

The Age variable is removed from the analysis in table 4 because the binary logistic regression didn’t show association between Age and STH at p- value ≤ 0.25. when age is categorized as 18-39 and ≥ 40 years

5. In table 2, no subjects under “Sometimes” for the washing hand hygiene. Better to delete them from the table.

• RESPONSE: 

o Deleted from table 2 (now relabeled as table 3) - line 229

6. Line 184-185: households who drained the liquids directly into the garden = 94.9%. It should be 94.7% (according to S2 file).

• RESPONSE: 

o corrected - line 224

7. Single infections were 83.8% in the abstract (line 32) and discussion (line 247) while 85.3% in the results (line 199) and 85.3% in line 311 in the discussion. Please check and correct.

• RESPONSE: 

o All are corrected as 85.3% - line 34, 192, 291 & 360

8. Line 192-195: no need for the CI to present the prevalence (%) of parasite detection.

• RESPONSE: 

o CI deleted - line 184

II: EDITING ISSUES:

• Abstract line 16: soil transmitted helminthes: change to “soil-transmitted helminths” (correct elsewhere in lines 41, 44).

• Corrected - line 16, 

• Line 41- 43 is removed to the adjust the word limit in the abstract to 300

Line 22: A community based: change to “A community-based” (correct elsewhere in line 97).

• Corrected - line 23

• Line 23: May to July, 2021: delete comma (similarly in line 99).

• Corrected - line 24 and line 102

• Line 30: A. lumbricoides: italic.

• Corrected : - line 32

• Line 31: T. trichiura: italic.

• Corrected - line 33

• Line 38: still remain: delete still (redundant).

• Deleted - line 40

• Line 60: major: change to “a major”.

• Corrected - line 62

• Line 68: socio- economic: remove space.

• Corrected - line 70

• Line 68: lack of accesses: change to “access”.

• Corrected - line 71

• Line 71: is: “are”.

• Corrected - line 73

• Line 98: population: change to “a population”.

• Corrected - line 101

• Line 102: temperature: change to “a temperature”.

• Corrected - line 106

• Line 129: Sufficient: change to “A sufficient”.

• Corrected - line 136

• Line 129: were collected: change to “was”.

• Corrected - line 136

• Line 130: leak proof: change to “leak-proof”.

• Corrected - line 137

• Line 131: unique Identifier (UID): change to “unique identifier” or “Unique Identifier” (UID).

• Corrected - line 138

• Line 133: specimen were: change to “specimen was”.

• Corrected - line 140

• Line 148: sold waste: change to “solid”.

• Corrected - line 224

• Line 156: was assess: change to “ assessed”.

• Corrected - line 168

• Line 182: About 56.1% the households: change to “of” the households.

• Corrected - line 222

• Line 196-197: S. mansoni, H. nana and E. vermicularis: full genera names.

• RESPONSE: - line 184

o Reviewer #1 recommend me to delete those parasite as it didn’t appear in table

• Line 199: had single infection: change to “a single”.

• Corrected - line 192

• Line 220: washing / peeling: remove spaces.

• Corrected - line 263

• Line 222: p- value: remove space

• Corrected - line 269

• Line 222: finger nail: change to fingernails (correct elsewhere in 224, 230, 231, 253, 374 and in tables 2 and table 4).

• All are corrected - line 36, 231, 264, 267, 273, 274, 284, 298, 391, 399 and 427

• Line 250: about 80.1% households: “of” households.

• Corrected - line 294

• Line 272: as public health problem: “a public”.

• Corrected - line 315

• Line 289: indirect: “an indirect”.

• Corrected- line 334

• Line 336: finding the Ethiopia Demographic: “of” the Ethiopia Demographic.

• Corrected- line 385

• Line 347: counter parts: one word.

• Corrected- line 397

• Line 374: soil- transmitted: remove space. 

• Corrected - line 428

• Table 1: N (%): change to “n” (%) (correct elsewhere for consistency).

• Table 3 and Table 4: No (%): change to “n” (%). 

• Corrected in all tables including the 

• Line 247: (88,5%): (88.5%).

• Corrected- line 291

• Line 308: (19.1%). were found: remove the full stop after the bracket.

• Corrected- line 357

• Line 309: Species: “species” (uncapitalize the word).

• Corrected - line 358

• Line 311: (83.1 %,): remove the comma after correcting the number (see comment 6).

• Corrected - line 360

• The AOR for untrimmed hand fingernail was 2.99, change the 2.9 (line 229) to 3. Similarly, wealth status (2.7) in line 232.

• Corrected both in text and table 4 - line 273, 275, 284

III: References:

Many references are not according to the journal style, please correct accordingly.

• RESPONSE: 

o All the references are corrected according to the recommended journal style(Vancouver)

o Additionally, New reference No= 27 and 28 explaining the detailed procedure recommended by the WHO was added to explain the concern of the reviewer 

Note:

• General

o All the comments and suggestion given by reviewer #1 and reviewer #2 are corrected

o Acknowledgement: the following sentenced was added to the acknowledgment “Dr. Ming-Chieh Lee mapped the study sites, for which we are grateful.” 

• Editorial and numerical errors 

o Some additional editorial, grammar, and spelling errors identified by the language experts were also corrected as indicated in the manuscript with truck changes.

o Some numerical errors identified by the authors are corrected, in table 1 and table 3 (now relabeled as table 2), the total number of hookworms identified was 9, not 8 and accordingly, prevalence figures under the column, Hookworm species in table 1 was changed (as indicted in truck changes) 

o In table 3 (now relabeled as table 2), the geometric mean under the column, hookworm species is corrected as 110.64( not 112.62)

• Figure 

o A new figure (Fig 1) (a map of the study sites) is added and labeled as figure 1. - line 111

o Fig 1 is relabeled as Fig 2. - line 123

o Fig 2 is relabeled as Fig 3.- Line 200

o Fig 2 is corrected as the number of hookworm species only, was 6, not 5, and the total number of hookworms was 9, not 8. The total denominator was 68, not 67, and hence the percentage in figure 2 (currently relabeled as figure 3) was corrected as follows:

---

## [Decision Letter · Decision Letter 1]

2 Sep 2022

Prevalence and intensity of soil-transmitted helminth infections and associated risk factors among household heads living in the peri-urban areas of Jimma town, Oromia, Ethiopia: A community-based cross-sectional study

PONE-D-22-15060R1

Dear Dr. Zeynudin,

We’re pleased to inform you that your manuscript has been judged scientifically suitable for publication and will be formally accepted for publication once it meets all outstanding technical requirements.

Kind regards,

Hesham

Hesham M. Al-Mekhlafi, PhD

Academic Editor

PLOS ONE

Additional Editor Comments (optional):

Reviewers' comments:

Reviewer's Responses to Questions

**Comments to the Author**

1. If the authors have adequately addressed your comments raised in a previous round of review and you feel that this manuscript is now acceptable for publication, you may indicate that here to bypass the “Comments to the Author” section, enter your conflict of interest statement in the “Confidential to Editor” section, and submit your "Accept" recommendation.

Reviewer #1: All comments have been addressed

Reviewer #2: All comments have been addressed

2. Is the manuscript technically sound, and do the data support the conclusions?

Reviewer #1: Yes

Reviewer #2: Yes

3. Has the statistical analysis been performed appropriately and rigorously? 

Reviewer #1: Yes

Reviewer #2: Yes

4. Have the authors made all data underlying the findings in their manuscript fully available?

Reviewer #1: Yes

Reviewer #2: Yes

5. Is the manuscript presented in an intelligible fashion and written in standard English?

Reviewer #1: Yes

Reviewer #2: Yes

6. Review Comments to the Author

Reviewer #1: The authors have satisfactorily responded and addressed all the comments raised in the last round. The manuscript is technically sound and of high quality. It can be accepted for publication as it stands.

Reviewer #2: The author(s) successfully responded to the reviewer 2 comments and coherently addressed all the points raised.

7. PLOS authors have the option to publish the peer review history of their article (what does this mean?). If published, this will include your full peer review and any attached files.

Reviewer #1: **Yes: **Prof Uwem Friday Ekpo

Reviewer #2: **Yes: **Wahib M. Atroosh

---

## [Editor Report · Acceptance letter]

6 Sep 2022

PONE-D-22-15060R1 

Prevalence and intensity of soil-transmitted helminth infections and associated risk factors among household heads living in the peri-urban areas of Jimma town, Oromia, Ethiopia: A community-based cross-sectional study 

Dear Dr. Zeynudin:

I'm pleased to inform you that your manuscript has been deemed suitable for publication in PLOS ONE. Congratulations! Your manuscript is now with our production department. 

Kind regards, 

on behalf of

Prof. Hesham M. Al-Mekhlafi 

Academic Editor

PLOS ONE